# Case Report of COVID-19 after Full Vaccination: Viral Loads and Anti-SARS-CoV-2 Antibodies

**DOI:** 10.3390/diagnostics11101815

**Published:** 2021-09-30

**Authors:** Magdalena Komiazyk, Jaroslaw Walory, Jan Gawor, Iza Ksiazek, Robert Gromadka, Anna Baraniak

**Affiliations:** 1Department of Drug Biotechnology and Bioinformatics, National Medicines Institute, 00-725 Warsaw, Poland; m.komiazyk@nil.gov.pl (M.K.); j.walory@nil.gov.pl (J.W.); i.ksiazek@nil.gov.pl (I.K.); 2DNA Sequencing and Synthesis Facility, Institute of Biochemistry and Biophysics, Polish Academy of Sciences, 02-106 Warsaw, Poland; gaworj@wp.pl (J.G.); robert@ibb.waw.pl (R.G.)

**Keywords:** mRNA vaccine, anti-SARS-CoV-2 antibodies, viral loads, COVID-19

## Abstract

The introduction of effective vaccines against SARS-CoV-2 is expected to prevent COVID-19. However, sporadic cases of infection in vaccinated persons have been reported. We describe a case of a double-dose vaccinated woman with COVID-19. All stages of infection were observed, from no identification of virus, then the start of the infection, a high viral load, coming out of viraemia, and finally no detection of the virus. Despite the high viral load, the woman demonstrated mild COVID-19 symptoms, manifested only by a sore throat. The antibody results showed that she produced both post-infectious and post-vaccination immune responses. Phylogenetic analysis of the obtained viral genome sequence indicated that the virus belonged to the UK SARS-CoV-2 lineage B.1.1.7 (GR 501Y.V1; 20I/S:501Y.V1; Alpha variant).

## 1. Introduction

The worldwide spread of a novel coronavirus called severe acute respiratory syndrome coronavirus 2 (SARS-CoV-2), which causes the severe upper respiratory tract infection, coronavirus disease 2019 (COVID-19), constitutes a huge challenge for public health today [1,2]. The detection of viral RNA by real-time reverse transcription quantitative polymerase chain reaction (RT-qPCR) is considered the gold standard for screening and diagnosing this infection [3,4]. New serological tests are helpful as a complementary COVID-19 diagnostic tool and/or may play a significant role in post-vaccine immune response studies [5,6]. Vaccination is one of the most effective means by which to prevent infectious diseases. Recently, various vaccines have become available to protect against COVID-19; four of them are currently registered for administration in countries of the European Union [7]. Although the currently approved COVID-19 vaccines effectively prevent serious COVID-19 illness and significantly reduce the rates of hospitalization, they do not confer 100% protection against SARS CoV-2 infection [8,9]. For this reason, several cases of infection in vaccinated individuals have already been reported [9,10,11]. However, data on the nature of breakthrough infections after COVID-19 vaccines are still lacking.

## 2. Case Description

A 54-year-old woman (Caucasian race) with three chronic diseases (hyperlipidemia since 1994, type 2 diabetes since 2008, and hypothyroidism since 2014) received the first dose of an mRNA vaccine, BNT162b2, (Pfizer, Philadelphia, PA, USA and BioNTech, Mainz, Germany) against SARS-CoV-2 on 24 January 2021. Nine days before (15 January), her serum was tested for IgA and IgG antibodies against spike (S) protein (IgA-S1 and IgG-S1) and IgG antibodies against nucleocapsid (N) protein (IgG-N) using three commercial serological assays: Anti-SARS-CoV-2 ELISA (IgA-S1), Anti-SARS-CoV-2 QuantiVac ELISA (IgG-S1), and Anti-SARS-CoV-2-NCP ELISA (IgG-N) (EUROIMMUN Medizinische Labordiagnostika AG, Lübeck, Germany). According to the manufacturer’s instructions, the obtained results were interpreted as negative: IgA-S1, 0.197 RU/mL; IgG-S1, 3.545 RU/mL; and IgG-N, 0.023 RU/mL. On 14 February, the woman received a second dose of the vaccine. Vaccinations were routinely administered by intramuscular injection into the deltoid muscle. No serious adverse vaccine reactions were observed; she experienced only redness at the injection site after both doses of vaccine.

On the weekend of 20–21 February, the woman’s unvaccinated husband started to manifest symptoms of infection. He tested positive for SARS-CoV-2 in an upper respiratory tract specimen by RT-qPCR on 22 February. The test was performed using a commercial MutaPLEX Coronavirus (SARS-COV-2) kit (Immundiagnostik AG, Bensheim, Germany) that detects three viral RNA fragments: SARS-CoV-2–specific spike (S gene) and RNA polymerase-dependent RNA (RdRP gene) regions, and envelope (E gene) region characteristic of both known SARS viruses. Furthermore, this assay contains a control RNA (internal process control, IPC) that allows the detection of RT-PCR inhibition and acts as control, in which the nucleic acid is isolated from the specimen. Additionally, the kit includes an internal system control (ISC), which enables the detection of a housekeeping gene (Beta-actin) from biological sample. The ISC helps avoid false negative results due to insufficient sampling. PCRs were carried out with the Applied Biosystems QuantStudio 6 Pro Real-Time PCR System (Life Technologies Holdings Pte Ltd, Singapore, Singapore) according to the manufacturer’s instruction. The number of gene copies per reaction mix was determined from the appropriate standard curve based on the cycle number at the set threshold fluorescent intensity. The husband presented a high viral load with ~10^5^ copies of the S/RdRP genes and ~10^6^ copies of the E gene in the sample.

Because the husband had home contact with his wife, the woman was also examined a day later (23 February); the result was negative. As she still remained in very close contact with her infected husband, despite being vaccinated, she was re-examined on 26 February. The test yielded an inconclusive result; specifically, the E gene was detected in the sample with a copy number less than ten. Over the weekend (27–28 February), the woman began to feel a sore throat that lasted for five days. She showed no other symptoms of COVID-19, such as fever, cough, shortness of breath, muscle or body pain, and anosmia. The next day (1 March) she tested positive for COVID-19 and demonstrated an increased viral load, with ~10^5^ copies of the S/RdRP genes and ~10^6^ copies of the E gene in the sample. The woman was isolated according to the current epidemiological recommendations [12]. After ten days (11 March, end of isolation), she was again tested and the result was inconclusive; only the E gene with less than ten copies was detected. On 15 March, she no longer had any detectable viral genes. The RT-qPCR results and timeline of the described case are shown in Table 1 and Figure 1a.

On 15 March, the woman was also tested for SARS-CoV-2 antibodies, as previously described. There were significant conversion levels for IgA-S1, 16.365 RU/mL, and IgG-S1, 295.841 RU/mL. A positive result for IgG-N with 2.308 RU/mL was obtained. The immune response was further monitored. On 14 May, IgA-S1 and IgG-S1 levels increased to 19.139 RU/mL and 2180.299 RU/mL, respectively. The IgG-N decreased to 1.541 RU/mL. On 18 August, a significant decrease in antibody levels was observed to 11.067 RU/mL, 696.692 RU/mL, and 0.731 RU/mL for IgA-S1, IgG-S1, and IgG-N, respectively. Dynamic changes in anti-SARS-CoV-2 antibody levels are shown in Figure 1b.

## 3. Identification of SARS-CoV-2 Variant

The virus variant known as UK SARS-CoV-2 lineage B.1.1.7 (GR 501Y.V1; 20I/S:501Y.V1; Alpha variant) [13] was identified by RT-qPCR using the Bosphore^®^ SARS-CoV-2 Variant Detection Kit v1 (Anatolia Geneworks, Istanbul, Turkey). In addition, viral genome sequencing was performed. The patient’s RNA sample was reverse transcribed using the LunaScript-RT SuperMix kit (New England Biolabs, Ipswich, MA, USA), followed by PCR amplification of ~2.5 kb products [14] that were further sequenced using SQK-LSK109 chemistry and an R94.1 flowcell on a GridION instrument (Oxford Nanopore Technologies, Oxford, UK). The ARTIC software pipeline (available online: https://artic.network/ncov-2019/, accessed on 23 March 2021) was used for data trimming, PCR primer removal, and genome consensus generation using the medaka polishing tool. Phylogenetic analysis of the genome sequence using Pangolin (available online: github.com/cov-lineages/pangolin accessed on 23 March 2021) confirmed the initial RT-qPCR result that the virus belongs to the B.1.1.7 lineage (Figure 2). The SARS-CoV-2 genome sequence has been deposited in the GISAID database [15] under the accession number EPI_ISL_1257898.

## 4. Discussion

The global spread of SARS-CoV-2 is a public health problem of great concern. This virus was initially identified in China in December 2019 and soon thereafter, in March 2020, the World Health Organization declared COVID-19 a pandemic [1,2]. The introduction of effective vaccines against SARS-CoV-2 is expected to prevent the incidence of this disease, ultimately leading to community protection [16]. The mRNA vaccines have shown high effectiveness in the prevention of COVID-19, severe disease hospitalization, and death [8,9]. The effectiveness in preventing after clinical trials for the BNT162b2 vaccine was 95% [17]. In a study from Israel, after a six-month post-vaccination follow-up in a randomized, placebo-controlled clinical trial, vaccine BNT162b2 was shown to be 91.3% effective in preventing COVID-19 disease [8]. As the majority of mRNA vaccines are based on the SARS-CoV-2 spike antigen, their effectiveness may therefore be affected by a change in antigen. In some studies involving different variants of SARS-CoV-2, the efficacy has been shown to be slightly lower than originally expected; for B.1.1.7 lineage over 70%, while for SARS-CoV-2 B.1.617.2 (Delta variant), the effectiveness varies between 50 and 70% [18,19]. These data demonstrate that monitoring vaccine efficacy for new variants is essential to pandemic control.

As mentioned earlier, some cases of infection in vaccinated persons have already been reported. Fully vaccinated individuals are still susceptible to SARS CoV-2 infection, especially if a partner, household member, or cohabitant refuses to be vaccinated. However, these infections are usually mild, often asymptomatic [8,9,10]. An important question is whether current vaccines will also prevent transmission of the virus. There is no report so far that the COVID-19 vaccine can do this.

We describe a case of a double-dose vaccinated woman who was confirmed to have SARS-CoV-2 after contact with her infected husband. All stages of infection were observed, from no identification of virus, then the start of the infection, a high viral load, coming out of viraemia, and finally no detection of the virus. Her antibody levels before vaccination showed that she had no prior SARS-CoV-2 infection. The level of anti-SARS-CoV-2 antibodies was not tested in the woman after the second dose of the vaccine, immediately before she contracted the disease. We cannot determine whether insufficient antibodies generated after vaccination contributed to the disease. However, the antibody results examined two weeks after infection and four weeks after vaccination showed that she generated both post-infection (a positive result for IgG-N) and post-vaccination (high levels of IgA-S1 and IgG-S1) immune responses. The distance between the second dose of the vaccine and exposure to the virus could be too short and she did not produce protective antibody levels. Despite comorbidities that would put the woman at risk of an unfavorable prognosis of infection, she had mild symptoms of a cold, manifested only by a sore throat. This is the additional purpose of the vaccination. We also investigated the level of her antibodies produced three and six months after the second dose of the vaccine. A decrease in IgG-N was observed; after six months their level was below that considered positive. The post-vaccination response (IgG-1S and IgA-S1), initially elevated in both antibody classes, indicates that the infection acts as a booster.

More than 90% of neutralizing antibodies from COVID-19 patients have been shown to target the virus spike receptor-binding domain (RBD) [20]. The B.1.1.7 variant identified in the woman has a N501Y mutation, which is located at the interaction surface between the RBD and human Angiotensin Converting Enzyme 2. This mutation has been shown to result in higher binding affinity of the virus to the cell receptor, which could account for the much higher infectious rate. However, this does not affect the binding of sera from vaccinated individuals [21]. Another study demonstrated a small reduction in neutralization against the B.1.1.7 variant in serum obtained from individuals vaccinated with two doses of BNT162b2 [21]. This variant also has a mutation in regions other than RBD, including deletions in the N-terminal domain (NTD), targeting NTD-specific neutralizing antibodies. Thus, it is possible that the infection, despite vaccination, can be related to the viral escape mechanism from NTD-specific neutralizing antibodies. However, this hypothesis requires additional verification.

Phylogenetic analysis of the obtained genome sequence (Figure 2) indicated that the virus belongs to the B.1.1.7 lineage, which is not surprising considering the current high prevalence (more than 60% of detected cases) of this variant at that time in Europe and worldwide [14]. Interestingly, BLAST genome analysis in the GISAID database revealed that the genome sequence shares 100% identity to five isolates from UK collected in mid-February and four virus isolates collected in Germany in mid-March.

## 5. Conclusions

Safe and effective prophylactic vaccines are essential to stop a pandemic with devastating medical, economic, and social consequences. However, despite vaccination, people may be infected with SARS-CoV-2 and present a high viral load without specific infection symptoms, thus becoming potential carriers of the virus. Therefore, we recommend using personal protective equipment for all vaccinated individuals. In the case of the appearance of any infection symptoms, test for the presence of the SARS-CoV-2 and apply isolation.

## Figures and Tables

**Figure 1 diagnostics-11-01815-f001:**
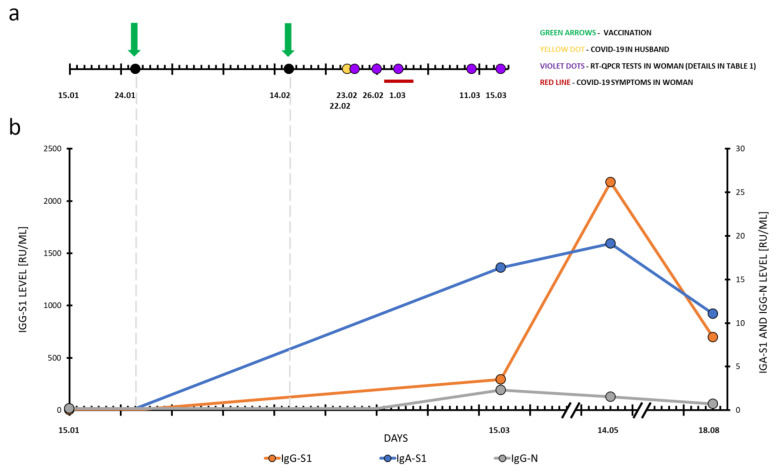
The timeline of the case. (a) shows the two doses of vaccine (green arrows), the identification of COVID-19 in the husband (yellow dot), the RT-qPCR tests performed in the woman (violet dots), and the duration of COVID-19 symptoms in the woman (red line). (b) shows the levels of anti-SARS-CoV-2 antibodies: IgG-S1 (orange line), IgA-s1 (blue line), IgG-N (grey line).

**Figure 2 diagnostics-11-01815-f002:**
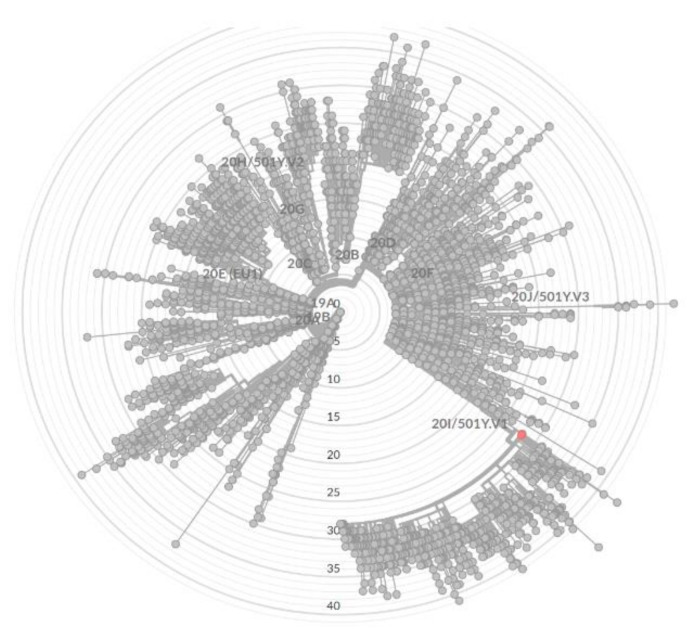
Phylogenetic analysis of SARS-CoV-2 isolate. Sample EPI_ISL_1257898 is marked as red dot.

**Table 1 diagnostics-11-01815-t001:** Results of RT-qPCR Analysis of Selected Genes during the SARS-CoV-2 Infection.

Data	Day	RdRp/S-Gene		E-Gene		ISC	
		Cq*	RNA Copies/Sample	Cq*	RNA Copies/Sample	Cq*	RNA Copies/Sample
23.02	1	Not detected	-	Not detected	-	25	2.5 × 10^3^
26.02	4	Not detected	-	36	<10	26	5 × 10^3^
1.03	7	21	10^5^	17	10^6^	21	10^5^
11.03	17	Not detected	-	36	<10	26	5 × 10^3^
15.03	21	Not detected	-	Not detected	-	24	10^4^

Cq*—Quantification cycles, the PCR cycle number at which a sample’s reaction curve intersects the threshold line.

## Data Availability

The SARS-CoV-2 genome sequence has been deposited in the GISAID database under the accession number EPI_ISL_1257898.

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
