# Peer review of "Case Report of COVID-19 after Full Vaccination: Viral Loads and Anti-SARS-CoV-2 Antibodies"

_diagnostics, 2021, doi:10.3390/diagnostics11101815_

Round 1

Reviewer 1 Report

  1. The authors carelessly state that " It is alarming that
    the woman presented COVID-19 despite receiving a double dose of the vaccine." However:
    1. The patient in the case report was tested/ contracted the virus after approximately 2-3 weeks of her second dose. A person is considered fully vaccinated after 2 weeks of the second dose of the Pfizer vaccine
    2. The vaccine is not 100% effective. It is supposed to be approx. 90-95% effectiveness which translates to thousands of fully vaccinated patients who will still contract the virus and show symptoms of disease.
  2.  The authors do mention that this patient could have worse clinical outcomes without the  vaccine, especially, with comorbidities. In my opinion, where the authors have reported the case as a tale of caution,  it should simply be reported as a case to aid other clinicians in determining the time of infection once exposed.
  3. Since the cause and time of contraction of infection is undetermined there is no evidence of vaccine failure in this patient case which is how the authors have presented the case report.
  4. I would caution that as scientists we understand the necessity of full vaccination throughout the planet and should process with humility and extremely good judgement when publishing results regarding Covid-19. This case report while soundly written, concludes that despite the vaccines, there are chances of infection contraction. Since there lack of discussion regarding the effectiveness of the vaccine as well as timing of the viral spread, the current discussion and conclusion are misleading to a lay reader interpreting the effectiveness of the vaccine.

Author Response

We thank the Reviewers for all their comments and recommendations. We rewrite our manuscript to meet the requirements of the Reviewers. The addition of new results supplemented and enriched our manuscript. Due to the new data we have included in the manuscript, we have decided to change the title to ‘Viral load and anti-SARS-CoV-2 antibodies monitoring in the vaccinated woman with COVID-19’.

Reviewer 2 Report

The authors' case report of SARS CoV-2 infection in a 54-year-old woman after receiving two dosages of COVID-19 mRNA vaccine (Pfizer, USA and BioNTech, Germany) i.e. “double vaccinated” contains thorough and systematic information.

The  case report illustrates the risk of SARS CoV-2 virus transmission from unvaccinated individual to  vaccinated individual. The information is important because it underscores the need of all members in a household to be educated about the consequences of one family member or co-habitant refusing vaccination or choosing not to wear face coverings when not vaccinated.   

The authors conclusion/discussion omit the fact that commercially approved COVID-19 vaccines are proven highly efficacious in inducing protective immune responses and unequivocally reduce the incidence of severe COVID-19 disease. The clinical trials and literature on this subject are extensive and must be cited in the references, in addition to the reports of SARS CoV-2 infection among vaccinated health care workers. By definition the outcomes of a single case report are of no value for the interpretation of vaccine efficacy or the benefit of neutralizing antibody responses.

Recommendations:

  1. The data presentation in case report can be vastly improved by a timeline showing dates of the two vaccine dosages, Cq values, antibody levels and COVID-19 symptoms. This can be presented either in a Figure or Table rather than a lengthy descriptive account of the data which is difficult for the reader to decipher.
  2. COVID-19 symptoms are incomplete. authors mention  "sore throat" The standard list of symptoms should be documented  (cough, fever, shortness of breath, difficulty breathing, loss of smell or taste, muscle or body pain, ansomnia)  
  3. Why is the UK  alpha variant in this case report relevant?  Is there any evidence in peer reviewed literature to indicate that GR 501Y.V1; Alpha variant viral load is associated with greater risk of infection among vaccinated individuals than other SARS-CoV-2 variants i.e. beta, delta, gamma variants or the S protein mutations in PANGO lineages?  

Comments:

pg 2,  line 46 anti not „anty“.

line 82. “there was a positive result for IgG-N, with 2.308 RU/ml”. At what time point? Are there follow up measurements? The anti-nucleocapsid (N)-protein antibodies are indicative of the natural SARS-CoV-2 infection. For this reason both the N-protein IgA and IgG levels they should be reported.

lines 116-120.  Conclusion is confused and inaccurate. The woman was likely not exposed to SARS CoV-2 infection because no antibodies were found in the reference serum prior to vaccination.   N-protein IgA and IgG levels reveal the exposure to natural infection. The S-protein IgG titers in Euroimmun Quantivac are relevant for predicting the neutralizing antibodies induced by vaccination and should be measured at the follow up after infection. The authors state it is “ unfortunate that antibodies were not measured immediately after the second vaccine dosage”. This point  is not so relevant. Most importantly the authors should compare N protein and S protein antibody levels at baseline, after the first dosage and follow up 1-6 months post onset of COVID-19 symptoms.

Discussion line 123 and 124 “ it is alarming that the woman presented COVID-19 despite receiving a double dose of the vaccine”. The duration of the symptoms and the time for resolving the infection should be mentioned in this context. It is noteworthy that influenza vaccines do not offer complete protection against influenza and vaccine effectiveness varies between H1N1 versus H3N2 subtypes.

Author Response

(The authors gave the same response as above.)

Reviewer 3 Report

In this article, the authors follow a double-dose vaccinated woman with of COVID-19 (Alpha variant of SARS-CoV-2) throughout all stages of infection, during which she demonstrated a high viral load but mild COVID-19 symptoms, and produced both post-infectious and post-vaccination immune responses.

The title of the article is straightforward and apposit, the keywords are well chosen, the abstract is clear and appropriate in lengh and content, but the last sentence seems a bit out of place, beeing a recommendation: I suggest to leave it in the conclusions but remove it from abstract.
The introdution is brief but sufficient to provide the necessary background. The references are relevant and recent.
The case description is clear and detailed. Maybe the viral load of the infecting husban may be reported (if known) because it can be of interest.
The table layout is clearly readable.
The novelty of this case, and therefore maybe the potential interest for the readers of the journal is not so high, but nevertheless the scientific soundness of this research is solid and so in my opinion it is, all considered, well suited for publication.

Author Response

We thank the Reviewer for the comments and recommendations.

Responses to Reviewer's suggestions:

In this article, the authors follow a double-dose vaccinated woman with of COVID-19 (Alpha variant of SARS-CoV-2) throughout all stages of infection, during which she demonstrated a high viral load but mild COVID-19 symptoms, and produced both post-infectious and post-vaccination immune responses.

  • The title of the article is straightforward and apposite, the keywords are well chosen, the abstract is clear and appropriate in length and content, but the last sentence seems a bit out of place, being a recommendation: I suggest to leave it in the conclusions but remove it from abstract.

Autor response:

This has been changed as suggested by the Reviewer.

  • The introduction is brief but sufficient to provide the necessary background. The references are relevant and recent. The case description is clear and detailed. Maybe the viral load of the infecting husband may be reported (if known) because it can be of interest.

Autor response:

This has been corrected as suggested by the Reviewer.

  • The table layout is clearly readable. The novelty of this case, and therefore maybe the potential interest for the readers of the journal is not so high, but nevertheless, the scientific soundness of this research is solid and so in my opinion it is, all considered, well suited for publication.

Autor response:

Thank you for this summary.

Round 2

Reviewer 1 Report

The authors have made acceptable revisions in the manuscript and addressed all the queries. I would recommend that the manuscript be accepted for publication in its current form.

Author Response

Thank you for your review.

Reviewer 2 Report

The manuscript requires complete overall on  grammar and syntax

Unfortunately the data presentation is even more confusing than the first draft submission. 

Title  should be simplified. 

"Case report of COVID-19 after full vaccination: viral loads and anti-SARS-CoV-2 antibodies". 

. i.e. "Man" change to husband or partner.   

The authors did not follow instructions in first review. Added Figure timeline. Script  too small and text difficult to decipher.

The data should be presented as follows :

TIMELINE in  large size font. The timeline story has two parts. A continuous timeline beginning 15- January and ending on 15-March. The large tick marks should indicate weeks and small tick mark days. A double hash mark thru the X axis should indicate the broken timeline which begins with antibodies measured in the REFERENCE serum taken on 15 January and  resumes with Euroimmun ELISA data points on 14-May and 18-Aug 

  1.   Continuous timeline should show the following  a) the period of  COVID-19 symptoms  of the partner or husband, and the period of COVID-19 symptoms of the vaccinated women. Both periods should be highlighted in bold color bars below the timeline. b) dates of  two vaccine dosages should  represented by bold ARROWS. c) the Cp values of  rT-PCR results can be inserted inside a box. The  Cp should be explained to be representative of viral load.  
  2. Broken timeline is the antibody responses. The authors were instructed in first submission to delete the descriptive text  of Euroimmun ELISA data and instead present in a GRAPHED timeline.  The x axis should be days post onset of COVID-19 symptoms. The y axis should trace the values of  anti- S protein IgA, anti-S protein IgG and the anti-N protein IgG at the tree time points;  15 January, 14-May and 18-Aug . Authors must use  Large symbols with connecting solid line or dotted line to indicate the IgA S , IgG S quant and IgG N.

 The important  take home message from this case report is that fully vaccinated individuals are still susceptible  to SARS CoV-2, especially if the   partner, household members,  or  cohabitants refuse vaccination. 

In this case it is noteworthy that the vaccinated women developed significant viral loads but their COVID-19  illness was mild and without complications. 

Author Response

Thank you for your review.

Round 3

Reviewer 2 Report

The manuscript syntax needs  editing.

Abstract:

Abstract: The introduction of effective vaccines against SARS-CoV-2 is expected to prevent COVID- 19. However, it should be noted that the vaccine does not provide 100% protection against this dis- 12
ease; sporadic cases of infection in vaccinated persons have already been reported. 

Once again, the authors have not accurately stated the key issue. 

Introduction

The sentences line 37-45 need revision.  

Introduction. It should be noted that the vaccine does not provide Although the currently approved COVID-19 vaccines effectively prevent serious COVID-19 illness and significantly reduce the rates of hospitalization (INSERT references) , they  do not confer 100% protection  against SARS  CoV-2 infection. 
For this reason; several cases of infection in vaccinated individuals have already been  reported [8, 9, 10]

Authors should avoid beginning sentences with "It". "It should be noted..(line 37-38), it should be  remembered  (line 143),  "It is possible "(line 161-162), It has been reported .. (line 164), "It is important to remember..." (line 194-195),   

Discussion (lines 132-136) Major re-writing & editing is needed. The following recent reference is recommended. 

The  BNT162b2 vaccine continues to show a favourable safety profile and to be highly efficacious.  At six months follow-up after vaccination a randomized placebo-controlled clinical trial of  44,165 participants 16 years of age or older and 2264 participants 12 to 15 years of age the vaccine efficacy was 91.3%  (95% confidence interval  89.0 to 93.2)  in preventing COVID-19  disease  [New Eng J Med, 2021, Sept 15 ].

Conclusion 

line 195

Delete "it is important to remember" 

Author Response

Thank you for your valuable comments on our manuscript. 
